Multiomics-based analyses of KPNA2 highlight its multiple potentials in hepatocellular carcinoma

Zhang Jinzhong 1
Zhang Xiuzhi 1 2
Wang Lingxiao 1
Kang Chunyan 1
Li Ningning 1
Xiao Zhefeng xiaozf@csu.edu.cn 2
Dai Liping lpdai@zzu.edu.cn 3
1 Department of Pathology, Henan Medical College , Zhengzhou , Henan Province , China
2 NHC Key Laboratory of Cancer Proteomics, Xiangya Hospital, Central South University , Changsha , Hunan Province , China
3 Henan Institute of Medical and Pharmaceutical Sciences, Zhengzhou University , Zhengzhou , Henan Province , China
Shang Yuan
Electronic publication date: 2021 Sep 21
Publication date: 2021
Volume: 9
Electronic Location ID: e12197
Received 2021 Jun 2; Accepted 2021 Aug 31
Copyright: ©2021 Zhang et al.
Copyright year: 2021
Copyright holder: Zhang et al.
License: This is an open access article distributed under the terms of the Creative Commons Attribution License, which permits unrestricted use, distribution, reproduction and adaptation in any medium and for any purpose provided that it is properly attributed. For attribution, the original author(s), title, publication source (PeerJ) and either DOI or URL of the article must be cited.
License URL: https://creativecommons.org/licenses/by/4.0/

Keywords: Hepatocellular carcinoma, KPNA2, Genetic and epigenetic regulation, Prognosis, Fatty acid metabolism, Immunoregulation

Funding: The Key Scientific Research Project of Colleges Universities in Henan Province of China 20B320007 Joint Construction Project of Henan Medical Science and Technology Research Plan LHGJ20190717 Science and Technology Project in Henan Province of China 142102310441 This work was funded by The Key Scientific Research Project of Colleges and Universities in Henan Province of China (grant number 20B320007), the Joint Construction Project of Henan Medical Science and Technology Research Plan (LHGJ20190717), and a grant from the Science and Technology Project in Henan Province of China (grant number 142102310441). The funders had no role in study design, data collection and analysis, decision to publish, or preparation of the manuscript.

==============================
Dysregulation and prognostic roles of Karyopherin α2 (KPNA2) were reported in many malignancies including hepatocellular carcinoma (HCC). A multi-omics analysis of KPNA2 is needed to gain a deeper understanding of its multilevel molecular characteristics and provide novel clues for HCC diagnosis, prognosis, and target therapy. Herein multi-omic alterations of KPNA2 were analyzed at genetic, epigenetic, transcript, and protein levels with evaluation of their relevance with clinicopathological features of HCC by integrative analyses. The significant correlations of KPNA2 expression with its gene copy number variation (CNV) and methylation status were shown through Spearman correlation analyses. With Cox regression, Kaplan-Meier survival, and receiver operating characteristic (ROC) analyses, based on the factors of KPNA2 CNV, methylation, expression, and tumor stage, risk models for HCC overall survival (OS) and disease-free survival (DFS) were constructed which could discriminate the 1-year, 3-year, and 5-year OS/DFS status effectively. With Microenvironment Cell Populations-counter (MCP-counter), the immune infiltrations of HCC samples were evaluated and their associations with KPNA2 were shown. KPNA2 expression in liver was found to be influenced by low fat diet and presented significant correlations with fatty acid metabolism and fatty acid synthase activity in HCC. KPNA2 was detected lowered in HCC patient’s plasma by enzyme linked immunosorbent assay (ELISA), consistent with its translocation to nuclei of HCC cells. In conclusion, KPNA2 multilevel dysregulation in HCC and its correlations with immune infiltration and the fatty acid metabolism pathway indicated its multiple roles in HCC. The clinicopathological significance of KPNA2 was highlighted through the in-depth analyses at multilevels.

Introduction

As the most commonly diagnosed malignant tumor in liver with a 5-year overall survival (OS) rate of below 20% after diagnosis, hepatocellular carcinoma (HCC) is one of the most aggressive and least understood cancers worldwide (Wang et al., 2016). The lethality of HCC is mainly due to delayed diagnosis, postoperative recurrence, and limited therapeutic options for advanced-stage HCC. It highlights an urgent need to clarify the potential molecular mechanisms underlying HCC development and develop promising clinical biomarkers (Farazi & De Pinho, 2006). Although increasing candidate biomarkers have been proposed to improve HCC diagnosis and prognosis (Singh, Kumar & Pandey, 2018), few enter into clinical application partly owing to the multiplex attributes and multifaceted roles of the biomarkers.

KPNA2, also known as importin α-1, is a member of the karyopherin α/importin α family. It participates in the classical nuclear protein import pathway as an adaptor binding the nuclear localization signal (NLS) of cargo proteins and linking them to importin β which ferries the ternary complex through the nuclear pore complex (Goldfarb et al., 2004). Elevated levels of KPNA2 have been observed in a variety of cancers including HCC and were correlated with poor prognosis in most solid tumors (Yoshitake et al., 2011; Zhou et al., 2017). Our pre-analysis of HCC gene expression data of public databases identified KPNA2 as a unique independent unfavorable predictor for HCC OS and demonstrated its increased expression in early stage of HCC (File S1), in accordance with other experimental evidence (Jiang et al., 2014; Yoshitake et al., 2011). KPNA2 was reported to enhance the proliferation and growth of HCC cells, reduce cellular apoptosis, and promote migration in vitro and in vivo (Guo et al., 2019). No nuclear expression of KPNA2 in non-tumorous liver tissues was observed while nuclear KPNA2 expression was significantly upregulated in HCC tissues, which was associated with a poor prognosis and a risk of recurrence in HCC patients (Jiang et al., 2014). It was reported that KPNA2 might promote HCC cell proliferation by increasing the expression of CCNB2/CDK1 (Gao et al., 2018). Another study suggested that KPNA2 regulated STMN1 by import of E2F1/TFDP1 in liver cancer (Drucker et al., 2019). Additionally, nuclear import of PLAG1 by KPNA2 was essential for the role of KPNA2 in HCC cells (Hu et al., 2014). HBV pre-S2 mutant large surface protein (LHBS) interacted with KPNA2 and blocked nuclear transport of an essential DNA repair and recombination factor NBS1, inducing genomic instability in HBV-infected hepatocytes and explaining the complex chromosome changes in HCCs (Hsieh et al., 2015). However, there still remains a rather unexplored field underlying KPNA2 molecular evidences and mechanisms in HCC.

In the present study, we performed a comprehensive and multilevel integrative analysis of KPNA2 to explore more signature modalities of KPNA2 in HCC including the genetic and epigenetic regulation of its expression, the landscape of its transcript/protein variants, and its association with immune infiltrations and fatty acid metabolism pathway in HCC. In addition, we constructed two risk models for HCC OS and disease-free survival (DFS) based on the multi-omic data of KPNA2 and evaluated their effectiveness. The laboratory work of testing plasma KPNA2 levels supported its nuclear translocation characteristic. The knowledge deepened the understanding of the deregulation and functioning of such an important biomarker in HCC and provided novel insights into HCC pathogenesis and treatment.

Material and Methods

Available data and data processing

The clinical features of The Cancer Genome Atlas (TCGA) HCC patients (TCGA-LIHC dataset, named TCGA-HCC in this study, n = 377) and their transcriptome profiling gene expression RNA-seq data (371 primary tumors, 2 recurrent tumors, and 50 paired normal liver specimens from 371 primary HCC patients) were downloaded from the genomic data common (GDC) portal (https://portal.gdc.cancer.gov/). The clinical features of the 377 patients were shown in Table 1. The expressional counts data TMM (trimmed mean of M-values) were normalized for further analyses and the relative expression data of KPNA2 were extracted. The copy number segments after removing germline copy number variation (370 primary tumors from the 377 patients) and the methylation illuminaMethy450 data (377 primary tumors and 50 paired normal liver specimens from the 377 patients) from TCGA-HCC dataset were downloaded from Xena UCSC browser (https://xenabrowser.net/), from which KPNA2′data were further extracted. The splice variants of KPNA2 were searched through the Ensembl genome database (http://mar2017.archive.ensembl.org/). The normalized RNA-seq data of the KPNA2 transcript variants in TCGA-HCC dataset were extracted from the TCGA Pan-Cancer gene expression data which was also downloaded from UCSC Xena browser. The mRNA profiles of the KPNA2 transcripts were analyzed between the paired normal and tumor samples of 50 HCC patients. To obtain the information of KPNA2 protein variants in HCC, Human Protein Atlas (HPA), a comprehensive antibody-based protein atlas that displays expression and localization patterns of proteins in a large portion of human tissues and organs (Uhlen et al., 2005), was searched.

Table 1 Clinical features of the 377 HCC patients from TCGA database.

Variables	Case, n (%)	
Age at diagnosis (yr.)		
<60	172 (45.6%)	
≥60	204 (54.1%)	
NA	1 (0.3%)	
Gender		
Male	255 (67.6%)	
Female	122 (32.4%)	
TNM stage		
I	175 (46.4%)	
II	87 (23.1%)	
III	86 (22.8%)	
IV	5 (1.3%)	
NA	24 (6.4%)	
Pathologic grade		
G1	55 (14.6%)	
G2	180 (47.7%)	
G3	124 (32.9%)	
G4	13 (3.4%)	
NA	5 (1.3%)	
Race		
White	187 (49.6%)	
Asian	161 (42.7%)	
Black or African American	17 (4.5%)	
American Indian or Alaska Native	2 (0.5%)	
NA	10 (2.7%)	
Survival status		
Alive	244 (64.7%)	
Dead	132 (35.0%)	
NA	1 (0.3%)	

Insights into the copy number variation, mutation, and methylation status of KPNA2 gene in HCC

The correlations of KPNA2 expression with the gene copy number variation (CNV) and methylation status in HCC were investigated by spearman correlation analyses. KPNA2 methylation values between HCC tumors and normal liver tissues were compared by Wilcoxon test. The age-, gender-, and tumor stage- corrected prognostic effects of KPNA2 CNV and KPNA2 methylation level were investigated through ezcox package (Wang et al., 2019) in R3.6.1 software. KPNA2 mutations in TCGA-HCC samples were investigated through the GDC portal (https://portal.gdc.cancer.gov/). Besides, the genes whose mutations might result in KPNA2 expression alteration were identified by the cancer biomarker/target discovery tool muTarget (https://www.mutarget.com/).

The variables (CNV and methylation) with significant prognostic effects as well as age, gender, tumor stage, and KPNA2 expression entered into multivariable cox regression analyses to construct the risk models for HCC OS and DFS. The risk model was set as follows: risk score= ∑i=1nβi∗expi

where n is the number of selected variables, β (i) is the coefficient of the variable i in the multivariable Cox regression analysis, and exp (i) is the gene expression, tumor stage, and KPNA2 CNV or methylation value of the CpG sites. According to the risk scores of the patients, high- and low- risk groups were distinguished according to the median risk score as the threshold. The survival differences between high- and low- risk patients were visualized via Kaplan-Meier survival analysis with log rank test. In the survival analysis, only the patients with OS time or DFS time >one month (30 days) were included. The effectiveness of the prognostic models was also evaluated in male patients and female patients, younger patients (age <= 60y) and older (age >60y) patients, early-stage (stage I and stage II) patients and late-stage (stage III and stage IV) patients respectively. For all the analyses, p < 0.05 was considered statistically significant.

Exploration of KPNA2 potential functions in HCC

To uncover the potential roles of KPNA2 in HCC, based on the tissue-specific protein-protein interaction (PPI) data from the DifferentialNet database (Basha et al., 2018), its liver-specific PPI network and Kyoto Encyclopedia of Genes and Genomes (KEGG) pathway enrichment of the interacting partners were investigated through NetworkAnalyst (https://www.networkanalyst.ca/) (Zhou et al., 2019). The DifferentialNet database provides a differential view of the human interactome by integrating current data of experimentally-detected PPIs with data of gene expression across tissues. The differential tissue-specific score of an interaction was computed as the difference between its tissue-specific score and its median score across all tissues. For the network construction, only the PPIs that score in the top 15 percentiles and bottom 15 percentiles were selected. To further explore KPNA2’s relationship with HCC immunoregulation, the microenvironmental infiltration of eight kinds of immune cells including T cells (CD3+T cells), CD8+ T cells, cytotoxic lymphocytes, NK cells, B lymphocytes, the cells originating from monocytes (monocytic lineage), myeloid dendritic cells, and neutrophils, plus two kinds of stroma cells (endothelial cells and fibroblasts) in TCGA-HCC patients were evaluated with Microenvironment Cell Populations-counter (MCP-counter) method (Becht et al., 2016). Their correlations with KPNA2 expression and HCC prognosis were investigated. In addition, KPNA2 expression in the immune infiltration cells including B cells, CD4+ T cells, CD8+ T cells, macrophages, and NK cells were compared between HCC and normal liver tissues in TCGA-HCC dataset via GEPIA (http://gepia2021.cancer-pku.cn/) with the one-way analysis of variance (ANOVA) function.

Investigation of KPNA2 association with fatty acid metabolism in HCC

As the liver plays crucial roles in fatty acid metabolism and dysregulations of fatty acids are related to fatty liver diseases and liver cancer (Barbier-Torres et al., 2020; Seo et al., 2020), we speculated that there might be connection between KPNA2 expression and fatty acid metabolism in liver. Herein, the effect of low fat diet on KPNA2 expression was evaluated in liver samples from eight individuals including four interference groups (with low-fat diet) and four control groups (with their habitual diet) in the GEO dataset GSE7117 using GEO2R (https://www.ncbi.nlm.nih.gov/geo/info/geo2r.html), an interactive web tool that allows users to compare two or more groups of samples in a GEO series. Furthermore, in consideration of the relationship of fatty acid dysregulation with HCC (Che et al., 2019; Montagner, Cam & Guillou, 2019), the correlations of KPNA2 with the genes of fatty acid metabolism pathway in TCGA-HCC primary tumors in KEGG database were investigated through Spearman correlation analysis with Hmisc package (https://CRAN.R-project.org/package=Hmisc) in R software. The potential implication of KPNA2 in fatty acid biosynthesis was evaluated from the purity-corrected Spearman correlations of KPNA2 with the 11 genes whose products were annotated to fatty acid synthase activity (http://amigo.geneontology.org/amigo/term/GO:0004312) via Tumor IMmune Estimation Resource (TIMER) (https://cistrome.shinyapps.io/timer/) and p < 0.05 was considered significant.

The correlations of KPNA2 with other genes in HCC were also investigated in the TCGA-HCC database. Gene set enrichment analysis (GSEA) was performed with the top positively and negatively KPNA2-correlated genes (Spearman’s correlation coefficient >0.3 or <−0.3) through a web-based gene set analysis toolkit (http://www.webgestalt.org/) to further uncover KPNA2-associated pathways and to see if fatty acid metabolism was in the enriched pathways.

Characterization of the variants of KPNA2 transcripts and proteins, and plasma testing

The differential expression of KPNA2 transcript variants between paired HCC and normal controls from the 50 HCC patients in the TCGA-HCC dataset was analyzed through paired sample Wilcoxon rank test and p < 0.05 was considered statistically significant. To further uncover the potential roles of KPNA2 transcripts in HCC development, the differentially expressed transcripts were evaluated to investigate their correlations with alpha fetoprotein (AFP) and albumin (ALB) gene expression of HCC samples by Spearman correlation analysis. Moreover, multivariable Cox regression analysis of the differentially expressed transcripts, with gender, age, and tumor stage as the covariates was performed to investigate the transcripts’ independent prognostic values.

For KPNA2 protein variants, the immunohistochemical staining patterns with the antibodies against different immunogenic fractions of KPNA2 in HCC and normal liver tissues were extracted from the HPA database and analyzed. The protein level of KPNA2 (full-length) in plasma was examined by enzyme linked immunosorbent assay (ELISA). The work involving the plasma specimens was reviewed and approved by the Ethics Committee of Xiangya Hospital of Central South University (approval number: 201801002). All samples were collected with informed consent in accordance with the Declaration of Helsinki. A total of 92 plasma samples with clinical information (51 samples from primary HCC patients and 41 samples from healthy individuals) were enrolled in the study (Table S1). The plasma samples were obtained between July 2018 and December 2018 from Xiangya Hospital of Central South University (Changsha, China) at the time of diagnosis before any therapy. KPNA2 concentration was measured in duplicate using commercially available ELISA kits (abx250558, Abbexa Ltd, UK) according to the manufacturer’s instructions. Absorbance was read at 450 nm with wavelength correction set at 540 nm using an ELISA plate reader (BioRad, CA, USA). Wilcoxon test was used for two-group comparison. ROC analysis was performed to investigate the diagnostic power of the factor with pROC package (Robin et al., 2011) in R software. For the analyses, p < 0.05 was considered to be statistically significant.

Results

Genetic and epigenetic regulations of KPNA2 and their prognostic value in HCC

By correlation analysis, a significant positive correlation of KPNA2 CNV with KPNA2 expression was shown in Fig. 1A (R = 0.43, p < 0.001). Four CpG sites locating in KPNA2 promoter including cg23206777, cg22429852, cg21018429, and cg21820889 presented negative correlations (p < 0.05, Figs. 1B–1E) with KPNA2 expression respectively in HCC, indicating their potential roles in regulating KPNA2 expression. All of the 13 CpG methylation sites in KPNA2 with location information, KPNA2 expression-correlation values, and p values were showed in Table S2. By comparison analysis of the four sites methylation between HCC tumors and the control groups, lower methylation of cg22429852 and higher methylation of cg23206777 was found in HCC than in the normal, while the other two sites methylation showed no significant difference (Figs. 1F–1I). It is noteworthy that the negative relationship of the four sites methylation with KPNA2 expression was conducted in the HCC data without normal liver data. Thus we deduced that certain methylation sites might be affected by negative feedback regulation under physiological and pathological processes, and cg22429852 might be the most critical methylation site for KPNA2 upregulation. For KPNA2 gene mutation inquiry in the GDC portal (https://portal.gdc.cancer.gov/), only four patients in TCGA-HCC dataset were found to have KPNA2 mutations and no significant KPNA2 expression difference was revealed between the HCC samples with and without KPNA2 mutation (p = 0.69, Fig. S1). As for other gene mutation effects on KPNA2 expression, by muTarget analysis with mutation prevalence set at 3%, five mutated genes appeared to be related to KPNA2 expression. As shown in Fig. S2, the mutations of TP53 (p < 0.01), DNAH10 (p < 0.01), TSC2 (p < 0.01), and RB1 (p < 0.01) were associated with KPNA2 upregulation while KPNA2 was more reduced in HCC samples with BAP1 mutations (p < 0.01) than in those without BAP1 mutations.

Figure 1 Correlations of KPNA2 CNV and KPNA2 methylation with KPNA2 expression in HCC.

(A) Significant positive correlation between KPNA2 CNV and KPNA2 expression. (B–E) Significant negative correlations between KPNA2 expression and methylation level of cg23206777, cg22429852, cg21018429, and cg21820889, respectively. (F) Higher methylation of cg23206777 on KPNA2 gene in HCC tumors than in the normal tissues. (G–H) No significant difference of methylation status of cg21018429 and cg21820889 on KPNA2 gene between HCC tumors and normal livers, respectively. (I) Lower methylation of cg22429852 on KPNA2 gene in HCC tumors than in the normal tissues. For (A–E), x-axis represented the relative expression of KPNA2 [log2(TMM+0.001)] in HCC samples while the y-axis indicated CNV or methylation level (beta value) of the CpG sites. For (F–I), x-axis represented samples of different groups and y-axis represented methylation value of KPNA2 gene. Spearman’s correlation analysis and Wilcoxon test were used for the analyses and p < 0.05 was considered significant. CNV, copy number variation; TMM, trimmed mean of M-values.

When adjusted with age, gender, and tumor stage, the methylation status of two CpG sites of cg23206777 (p = 0.026) and cg17985418 (p = 0.00258) was shown to have a favorable or an unfavorable prognostic effect on HCC OS respectively, while KPNA2 CNV and other KPNA2 CpG sites presented no significant correlation with HCC OS (Table S3). For evaluation of the CNV and CpG associations with HCC DFS (Table S4), KPNA2 CNV (p = 0.002) and the methylation value of cg14898140 (p = 0.006) were shown to have unfavorable prognostic effects on HCC DFS independent of age, gender, and tumor stage while no significant correlation of other CpG sites was shown.

Additionally, the risk models for HCC OS and DFS were constructed by multivariable Cox regression analysis individually. As shown in Fig. 2A, tumor stage, KPNA2 expression, and cg17985418 methylation were shown to be independent prognostic factors for HCC OS and the risk model was as follows: Risk scoreOS=0.40271∗tumor stage+5.49197∗cg17985418+0.55453∗KPNA2expression.

There was a shorter OS for the HCC patients with high-risk scores than the ones with low-risk scores (Fig. 2B). By ROC analysis, an area under the curve (AUC) of 0.814 for 1-year survival, 0.788 for 3-year, and 0.733 for 5-year survival was shown respectively in Fig. 2C, indicating the effectiveness of the risk model for predicting HCC patients OS. To compare the predictive efficiency for HCC OS between the constitutive risk model and KPNA2 expression alone, the predictive efficiency of KPNA2 for HCC OS was calculated. By comparison, although there was no significant difference in their AUCs of 1-year OS (0.814 vs.0.779, p = 0.113), the AUCs of 3-year (0.788 vs. 0.707, p = 0.003) and 5-year OS (0.733 vs. 0.665, p = 0.034) of the constitutive model were significantly higher than those of KPNA2 expression alone (Fig. S3). It indicated the superiority of a multi-variable risk score model in the prediction of HCC prognosis, especially for the long survival status of the patients.

Figure 2 Construction and evaluation of HCC OS risk model with KPNA2 data and clinical features.

(A) Tumor stage, KPNA2 expression, and cg17985418 methylation were independent prognostic factors for HCC OS and they were selected for the risk model construction. (B) According to the risk model, the HCC patients with high-risk scores presented shorter OS than the ones with low-risk scores. (C) The OS risk model could discriminate the 1-year, 3-year, and 5-year survival status of the HCC patients effectively. Multivariable Cox regression analysis was used for the risk model construction. Kaplan-Meier survival analysis with log rank test was used for the survival comparison between high and low risk groups. ROC analysis was used to evaluated the effectiveness of the OS risk model in OS status prediction. OS, overall survival; ROC, Receiver operating characteristic. For all the analyses, p < 0.05 was considered significant.

With regards to HCC DFS (Fig. 3), tumor stage, KPNA2 CNV, and cg14898140 methylation were presented to be independent prognostic factors for HCC patients, while the prognostic value of KPNA2 expression was not significant (Fig. 3A). The risk model was established as: Risk scoreDFS=0.489∗tumor stage+0.644∗KPNA2CNV+11.3∗cg14898140.

The HCC patients with high-risk scores had a shorter disease-free time than the patients with low-risk scores (Fig. 3B). From the ROC analysis, the AUCs for 1-year, 3-year, and 5-year HCC DFS was 0.721, 0.670, and 0.666 respectively (Fig. 3C).

Figure 3 Construction and evaluation of HCC DFS risk model with KPNA2 data and clinical features.

(A) Tumor stage, KPNA2 CNV, and cg14898140 methylation were independent prognostic factors for HCC DFS and they were selected for the risk model construction. (B) According to the DFS risk model, the HCC patients with high risk scores presented shorter disease-free time than the ones with low risk scores. (C) The DFS risk model could discriminate the 1-year, 3-year, and 5-year DFS status of the HCC patients effectively. Multivariable Cox regression analysis was used for the risk model construction. Kaplan-Meier survival analysis with log rank test was used for the survival comparison between high and low risk groups. ROC analysis was used to evaluated the effectiveness of the DFS risk model in DFS status prediction. DFS, disease-free survival; CNV, copy number variation; ROC, Receiver operating characteristic. For all the analyses, p < 0.05 was considered significant.

Figure 4 Independent prognostic effects of the OS and DFS risk models in HCC patients.

(A–B) Compared with low-risk scores, high risk scores indicated shorter OS regardless of gender. (C–D) Compared with low-risk scores, high risk scores indicated shorter OS regardless of age. (E–F) Compared with low-risk scores, high risk scores indicated shorter OS regardless of HCC stage. (G–H) Compared with low-risk scores, high risk scores indicated shorter disease-free time regardless of gender. (I–J) Compared with low-risk scores, high risk scores indicated shorter disease-free time regardless of age. (K–L) Compared with low-risk scores, high risk scores indicated shorter disease-free regardless of HCC stage. OS, overall survival; DFS, disease-free survival. Kaplan-Meier survival analysis with log rank test was used for survival analysis and p < 0.05 was considered significant.

Furthermore, the unfavorable prognostic effects of OS high-risk scores were shown in male HCC patients (p < 0.0001, Fig. 4A), female HCC patients (p <0.0001, Fig. 4B), young HCC patients (p < 0.0001, Fig. 4C), elderly HCC patients (p <0.0001, Fig. 4D), early-stage HCC patients (p < 0.0001, Fig. 4E), and late-stage HCC patients (p = 0.00026, Fig. 4F) individually, indicating that the OS risk score was an independent prognostic factor. Likewise, the HCC patients with DFS high-risk scores had shorter disease-free time than those with the low-risk scores in different gender groups (p < 0.01, Figs. 4G–4H), different age groups (p <0.0001, Figs. 4I–4J), and the early-stage group (p <0.0001, Fig. 4K). In the late-stage group (Fig. 4L), though only one of the 77 late-stage patients was grouped into the low-risk group, it still showed a longer disease-free time than most of the patients with high-risk scores. It indicated that the prognostic effect of the risk scores for HCC DFS was independent of gender, age, and tumor stage.

Immunoregulatory roles of KPNA2 in HCC

Via NetworkAnalyst, the liver-specific PPI network of KPNA2 was investigated and 68 proteins were found to have interactions with KPNA2 (Fig. 5A). By KEGG pathway analysis, the interacting proteins were enriched into five significant terms including viral carcinogenesis, focal adhesion, leukocyte transendothelial migration, PI3K-Akt signaling pathway, and adherens junction (Fig. 5B), suggesting the association of KPNA2 with HCC immunology.

Figure 5 Liver specific PPI network of KPNA2 and the correlation of KPNA2 with immune response.

(A) The PPI network of KPNA2 in liver. The blue dots genes participated in the pathways in (B). (B) The KEGG pathways that the genes in KPNA2 PPI network were enriched. (C–G) The significant positive correlations of KPNA2 expression with infiltration of monocytic lineage cells, T cells, B lineage cells, CD8+ T cells, and myeloid dendritic cells, respectively. (H–J) No significant correlation between KPNA2 expression and infiltration of cytotoxic lymphocytes, neutrophils, and NK cells. (K) The significant negative correlation of KPNA2 with infiltration of endothelial cells. (L) No significant correlation between KPNA2 expression and infiltration of fibroblasts. The red node in (A) represented KPNA2 protein, the black and blue nodes were the proteins which had liver-specific PPIs with KPNA2. The blue nodes also indicated the proteins significantly enriched in the KEGG pathways in (B). PPI, protein-protein interaction. Spearman correlation analysis was used and p < 0.05 was considered significant.

Further study revealed that KPNA2 expression was positively correlated with infiltration of five kinds of immune cells in HCC including monocytic lineage cells (R = 0.35, p = 4e −12, Fig. 5C), T cells (R = 0.23, p = 1e −05, Fig. 5D), B lineage cells (R = 0.14, p = 0.0071, Fig. 5E), CD8+ T cells (R = 0.14, p = 0.006, Fig. 5F), and myeloid dendritic cells (R = 0.13, p = 0.014, Fig. 5G). While no significant correlation of KPNA2 expression with cytotoxic lymphocytes (p >0.05, Fig. 5H), neutrophils (p >0.05, Fig. 5I), and NK cells (p >0.05, Fig. 5J) was indicated. As to the two kinds of stroma cells, a negative relationship between endothelial cell infiltration and KPNA2 expression was presented (R =  − 0.19, p = 0.00033, Fig. 5K), while no significant correlation was shown between KPNA2 expression and the level of fibroblasts (p > 0.05, Fig. 5L). The correlation of KPNA2 with immune cell infiltrations in early (TNM stage I) and middle-late stage (TNM stage II/III/IV) were evaluated separately (Fig. S4). Interestingly, the Spearman correlations of the five types of immune infiltration cells with KPNA2 appeared statistically significant or more significant in HCC middle-late stage than in the early stage. It indicated that KPNA2′  connection with HCC immune infiltration is becoming increasingly apparent during HCC progress. By Cox regression analysis, among the KPNA2-correlated immune cells and stroma cells, monocytic lineage cells and myeloid dendritic cells were associated with HCC OS (p < 0.05, Fig. 6A), while none presented prognostic significance for HCC DFS (p > 0.05, Fig. 6B).

Figure 6 Prognostic value of KPNA2-correlated immune and stroma cells in HCC.

(A) The gender-, age-, and tumor stage-corrected prognostic value of KPNA2-correlated immune and stroma cells for HCC OS. (B) The gender-, age-, and tumor stage-corrected prognostic value of KPNA2-correlated immune and stroma cells for HCC DFS. OS, overall survival; DFS, disease-free survival. Cox regression analysis with ezcox package in R software was used for survival analysis and p < 0.05 was considered significant.

GEPIA2021 analysis resulted that among the five kinds of immune cells including B cells, CD4+ T cells, CD8+ T cells, macrophages, and NK cells, KPNA2 expression was lower in CD4+T cells (p = 0.02) and macrophages (p < 0.01) in HCC tumors in comparison with that in normal liver tissues, as shown in Fig. 7. Considering its important functions in normal cells (Goldfarb et al., 2004), KPNA2 dysregulation might connect to the dysfunction of the immune cells in HCC.

Figure 7 KPNA2 expression comparisons in immune cells between HCC samples and their paired normal livers.

One-way ANOVA analysis was used for comparisons and p < 0.05 was considered significant.

Associations between KPNA2 expression and fatty acid metabolism

As shown in Fig. 8A, following a low-fat hypocaloric diet for 8 weeks, there was a significant decrease of KPNA2 expression (probes: 201088_at and 211762_s_at from GPL570) in the four obese liver tissues (p < 0.05), indicating a negative effect of low-fat diet on KPNA2 expression in liver.

Figure 8 Associations between KPNA2 expression and fatty acid metabolism in liver and HCC.

(A) Significant decreases of KPNA2 expression in the obese livers with low-fat diet in contrast to the controls. (B) Significant negative correlations between KPNA2 expression and the expression of the genes in KEGG fatty acid metabolism pathway. (C) Functional enrichment of KPNA2 negatively-correlated genes in KEGG fatty acid metabolism pathway in HCC. For (A–B), GEO2R tool and Spearman correlation analysis was used respectively and p < 0.05 was considered significant. For (C), the analysis was performed via Metascape (https://metascape.org) and the top 15 items were listed.

For the association of KPNA2 expression with fatty acid metabolism related genes in HCC, among the 42 genes in KEGG fatty acid metabolism pathway, 88.1% of them (37/42) were shown to be significantly and negatively correlated with KPNA2 expression (Table S5). The top ten negatively correlated genes were shown in Fig. 8B. When the KPNA2-negatively-correlated genes in fatty acid metabolism pathway were applied to functional enrichment analyses, multiple fatty acids metabolic processes were revealed including fatty acid degradation and beta-oxidation on the top (Fig. 8C). As the down-regulation of fatty acid degradation and beta-oxidation might lead to the increase of fatty acid in the liver, a positive correlation between KPNA2 expression and fatty acid level in HCC was deducible. In fact, as shown in Table 2, nine of the eleven genes (81.8%) related to fatty acid synthase activity were shown to be significantly positively correlated with KPNA2 expression in HCC. All of these results were consistent with the association between low fat diet and KPNA2 decrease in the liver from the beginning analysis.

In addition, the correlations of KPNA2 with other genes in HCC were investigated in the TCGA-HCC database. Its top positively and negatively correlated genes with Spearman’s correlation coefficient above 0.3 or below −0.3 were extracted (File S3). In KEGG GSEA result, KPNA2-correlated genes were most positively enriched in cell cycle pathway while negatively associated with a variety of metabolic pathways including fatty acid degradation (Fig. S5), consistent with KPNA2 negative involvement in fatty acid degradation in HCC.

The landscape of KPNA2 transcripts and protein variants in HCC

KPNA2 is encoded by the KPNA2 gene on the chromosome 17 which contains 11 exons spanning approximately 11 kb in the human genome. The full-length KPNA2 protein (UniProt_P52292) consists of 529 amino acids and weighs around 58 kDa, comprising an N-terminal hydrophilic importin β-binding domain (IBB, from 2nd to 60th amino acid), a central hydrophobic region which binds the cargo’s NLS (from 142nd to 403rd amino acid), and a short acidic C-terminus with no reported function. KPNA2 has seven splice transcript variants as shown in Fig. S6A). Both of ENST00000330459 (amount percentage range: 25.87%–100%) and ENST00000537025 (amount percentage range: 0−1.16%) code for the full length of KPNA2 protein, the former of which is the predominant transcript among all the variants (Fig. S6B)). ENST00000579754 and ENST00000584026 are derived from alternative splicing discarding the rear half of exon 5 and the posterior exons, resulting in two short protein fragments of N-terminal 143 and 134 amino acids respectively, which retain only the IBB domains. ENST00000583392 and ENST00000582898 are two non-coding RNAs (ncRNAs).

Paired samples Wilcoxon tests revealed that the two long coding transcripts (ENST00000330459 and ENST00000537025) (p < 0.05, Figs. 9A–9B) and the two ncRNAs (ENST00000583392 and ENST00000582898) (p < 0.05 Figs. 9F–9G) were higher expressed in HCC than in the normal controls. No significant expression difference of the two short protein-coding mRNA (ENST00000579754 and ENST00000584026) (p > 0.05, Figs. 9C–9D) and the nonsense mediated decay ENST00000583269 (p > 0.05, Fig. 9E) was shown between the paired normal liver and HCC samples. Among the four differentially expressed transcripts (Table S6), ENST00000330459 (r = 0.223, p = 1.583E-05) and ENST00000582898 (r = 0.189, p = 2.538E-04) were shown to be positively correlated with AFP expression. In contrast, the negative correlations of ENST00000330459 (r =  − 0.280, p = 4.384E-08) and ENST00000583392 (r =  − 0.147, p = 0.005) with ALB expression were shown. Previous studies supported that serum AFP level was positively associated with HCC nuclear KPNA2 expression (Jiang et al., 2014; Yoshitake et al., 2011). It substantiated KPNA2 involvement in liver dysfunction and HCC development.

Table 2 Correlations between KPNA2 and the genes related to fatty acid synthase activity in HCC.

Genes related to fatty acid synthase activity	Correlation	P value	
ELOVL1	0.597	9.816E−35**	
ELOVL2	0.219	4.046E−05**	
ELOVL3	0.397	1.806E−14**	
ELOVL4	0.380	2.776E−13**	
ELOVL5	0.403	6.207E−15**	
ELOVL6	0.161	2.776E−03**	
ELOVL7	0.475	8.444E−21**	
FASN	0.338	1.160E−10**	
MCAT	0.068	0.209	
OLAH	0.013	0.808	
OXSM	0.134	0.013*	
Notes.

HCC, hepatocellular carcinoma

* p < 0.05.

** p < 0.01.

For the analyses, the tumor purity-corrected Spearman correlation was evaluated and p < 0.05 was considered significant.

Figure 9 Expression differences of KPNA2 various transcripts between HCC and the paired normal liver controls.

(A–B) ENST00000330459 and ENST00000537025 were higher expressed in HCC than in the controls. (C–E) No significant expression differences of ENST00000579754, ENST00000584026, and ENST00000583269 between HCC and the controls. (F–G) ENST00000583392 and ENST00000582898 were higher expressed in HCC than the controls. Paired samples Wilcoxon test was used for expression comparisons and p < 0.05 was considered significant.

When the four differentially expressed transcripts were applied to multivariable Cox regression analysis with gender, age, and tumor stage as variates, tumor stage and the full-length KPNA2 coding transcript ENST00000330459 independently showed unfavorable prognostic significance for HCC OS (Fig. 10A) and DFS (Fig. 10B). While the other full-length KPNA2 coding transcript ENST00000537025, which made up a tiny constitution of the transcript variants, showed a significant favorable prognostic effect on HCC OS (HR = 0.91, p < 0.05, Fig. 10A).

Figure 10 Prognostic value of KPNA2 transcripts in HCC.

(A) The significant prognostic value of ENST00000330459, ENST00000537025, and tumor stage for HCC OS. (B) The significant prognostic value of ENST00000330459 and tumor stage for HCC DFS. OS, overall survival; DFS, disease-free survival. Multivariable Cox regression analysis was used for survival analysis and p < 0.05 was considered significant.

Although KPNA2 was reported to be upregulated in HCC tissues in a recent proteomic study (Jiang et al., 2019), the distribution of KPNA2 protein variants was unclear. Herein, the immunohistochemical staining patterns of KPNA2 variants between HCC and normal liver samples were extracted from HPA database. Two types of KPNA2 antibodies against different immunogenic fragments of KPNA2 resulted in two different staining patterns. Raised against the amino acids 480-529 in KPNA2 C-terminal, CAB015460 antibody reacted solely to the full length KPNA2 (UniProt_P52292), while HPA041270 antibody which was against the immunogenic sequence in N-terminal of KPNA2 detected all the KPNA2 protein variants including the full length KPNA2 and the other two short protein fragments (UniProt_J3KS65 and J3QLL0) which keep N-terminal domain. KPNA2 was stained by CAB015460 moderately in the cytoplasm/membrane with no nuclear staining in all of the three normal liver samples, while showed nuclear staining in all of the six HCC samples which ranged from weak to strong with rare cytoplasmic or membranous staining observed. The representative staining images were showed in Figs. 11A and 11B respectively. With the other antibody HPA041270 which indiscriminately detected all the KPNA2 protein variants, the three normal liver samples were extensively stained in both cytoplasm/membrane and nuclei (Fig. 11C). The staining differences of KPNA2 between using the two antibodies in the normal liver samples indicated that KPNA2 in the normal nuclei might be the short KPNA2 protein fragments which could only bind with antibody HPA041270, not CAB015460. The short fragments of KPNA2 might play physiological roles in normal liver cells. When HPA041270 was applied to seven HCC specimens, KPNA2 remained positive staining in cytoplasm, membrane, and nuclei in most samples. In the three HCC samples, KPNA2 was more intensive in HCC nuclei (Fig. 11D) than in normal nuclei (Fig. 11C), indicating that at least a part of KPNA2 variants were translocated to nuclei if not all, and it was most likely the full-length KPNA2 (UniProt_P52292).

Figure 11 Immunohistochemical staining of KPNA2 in HCC and normal liver tissues from HPA database.

(A) With the antibody CAB015460, KPNA2 was positive in cytoplasm/membrane of normal liver cells with negative nuclear staining. (B) KPNA2 was positive in nuclei of HCC cells (with the antibody CAB015460). (C) With the antibody HPA041270 , KPNA2 was extensively stained in both cytoplasm/membrane and nuclei of normal liver cells. (D) The positive nuclear staining became more intensive in HCC tumor cells in comparison with (C) (with the antibody HPA041270). All the visual fields were extracted from same magnification and the scale plates were shown.

We tested the full-length KPNA2 protein level in plasma. As shown in Fig. 12A, plasma KPNA2 (UniProt_P52292) was shown to be lower in HCC patients than in normal individuals (p < 0.001), which might be partly due to the translocation of KPNA2 to nuclei in HCC as described above. By ROC analysis, the AUC for plasma KPNA2 was 0.787 in discriminating HCC from normal controls (Fig. 12B). With an optimal cutoff value of 7.923 ng/ml, 65.9% (sensitivity) of the 51 HCC patients and 98.0% (specificity) of the 41 normal controls could be accurately discriminated, indicating the diagnostic power of plasma KPNA2 in HCC.

Figure 12 Plasma KPNA2 level and its discriminating power between HCC patients and healthy controls.

(A) Plasma KPNA2 concentration in HCC patients was lower than in healthy individuals by ELISA. (B) ROC curve analysis of plasma KPNA2 in discriminating HCC patients from healthy individuals. ROC, receiver operating characteristic; p < 0.05 was considered to be statistically significant.

Discussion

In our preliminary bioinformatics analysis on public datasets of HCC, KPNA2 was highlighted to be a unique independent predictor for poor OS of HCC among the differentially expressed genes, and the other independent indicator was tumor stage (File S1). Although the association with aggressive clinical characteristics of HCC indicated KPNA2 as a potential therapeutic target, targeting of the factor would probably be complicated by its molecular characteristics, dysregulation mechanisms, and multiple cellular processes that are associated with KPNA2.

KPNA2 deregulation mechanisms have been studied at transcriptional and post-transcriptional levels involving transcriptional regulation factors, miRNAs, and long non-coding RNAs (Feng et al., 2016; Xiang et al., 2019; Zan et al., 2019). Our study provided a supplementary viewpoint from genetic and epigenetic perspectives. Chromosomal instability including frequent chromosome gains (1q, 5, 6p, 7, 8q, 17q, and 20) and losses (1p, 4q, 6q, 8p, 13q, 16, 17p and 21) was usually described in HCC (Guichard et al., 2012). KPNA2 gene which is located at chromosome 17q has not been noted for its copy number alteration in HCC. Herein a high frequency of KPNA2 CNV gains was found in HCC and significantly associated with KPNA2 expression, indicating that KPNA2 gene CNV was a crucial component of its deregulation mechanisms. The four CpG sites with negative methylation-expression correlation with KPNA2 provided another potential regulatory mechanism for KPNA2 expression. In addition, KPNA2 CNV and methylation status presented as potential biomarkers for HCC OS/DFS prognosis. It is noteworthy that neither the CpG site cg17985418 nor cg14898140, a constitute part of the risk model for HCC OS or DFS with an unfavorable prognostic effect respectively, showed any significant relationship with KPNA2 expression. It might be explained in that certain CpG sites methylation might represent as DNA methylation patterns and specific features of three-dimensional genome architecture which could have an even more important effect on global gene expression than gene silencing by promoter methylation. At this point, the constitutive risk models involving CpG sites methylation are presumably more predictive for HCC prognosis than KPNA2 expression level alone, which has been calculated and verified in the Result section.

Most studies have focused on identifying the cargo proteins of which KPNA2 mediated nuclear localization. Both oncoproteins and tumor suppressor proteins are nuclear translocated with KPNA2 such as E2F1, OCT4, c-Myc, p53, p27, MDC1, FGF1/2, LEF-1, CHK1, BRCA1, S100A2, S100A6, RECQL, RAC1, p65, JNK1, STAT3, c-Jun, NBS1, and TBP-2 (Han & Wang, 2020; Martinez-Olivera et al., 2018; Zhou et al., 2021). These downstream signaling pathways including PI3K/AKT, P53, MAPK, cell cycle, Wnt/ β-catenin, and EMT-related pathways affect the growth, invasion, and metastasis of tumors (Han & Wang, 2020). Given that KPNA2 binds to oncoproteins and tumor suppressor proteins indistinctively, it might depend on oncogenic or tumor suppressive signaling and specific space-temporal contexts to distinctively regulate the transport of oncoproteins and tumor suppressor proteins. For instance, KPNA2 was one of candidate target genes of p53 which inhibits KPNA2 in liver cancer to suppress tumor cell growth (Winkler et al., 2014). Moreover, the fragmental KPNA2 variants which lack the NLS-binding domain but retain their ability to bind importin β might affect the full-length KPNA2’s function through competitively binding to importin β. KPNA2 could affect carcinogenesis by mistaken translocation of cancer-associated cargo proteins such as c-Myc (Duan et al., 2020) and PLAG1 (Hu et al., 2014), both of which could be found in the liver-specific KPNA2 PPI network. The nuclear import of PLAG1 by KPNA2 was reported to lead to the enhanced proliferation and invasive activities of HCC cells (Hu et al., 2014). Though KPNA2 has been revealed to bring a variety of malignant characteristics to HCC, its role in HCC immunology was seldom referred to.

Liver is a central immunomodulator that ensures organ and systemic protection while maintaining immunotolerance (Ringelhan et al., 2018). Persistent deregulation of the tightly controlled immunological network leads to necroinflammation, promotion of liver fibrosis, and subsequently the induction of tumorigenesis (Ringelhan et al., 2018). The bioinformatics evidence herein demonstrated the involvement of KPNA2 in HCC immunological network and a positive correlation with five types of HCC-infiltrating immune cells in agreement with the breakdown of liver immunotolerance. By virtue of KPNA2’s role in nucleocytoplasmic transport, a possible related mechanism is altered nuclear shuttling of immunological modulators. For instance, NLRP3, a pro-inflammatory molecule involved in inflammasome activation, was recently found to be an interaction partner for KPNA2 (Park et al., 2019), implying the participation of KPNA2 in an inflammatory cascade. On the other hand, KPNA2 expression was up-regulated in response to extracellular immune signals such as TNF α, IL-1 β, and IL-10 (Liu et al., 2015; Tao et al., 2015; Zhang et al., 2019), indicating the interaction of KPNA2 and an inflammatory microenvironment.

Liver is also crucial in fatty acid metabolism (Bhushan et al., 2019; Ordóñez et al., 2019). The negative relationships of KPNA2 with fatty acid degradation and beta-oxidation processes and its positive involvement in fatty acid synthase activity demonstrated KPNA2’s role in dysregulation of fatty acid metabolism. Considering low-fat diet effect on KPNA2 decrease in obese livers, diet control might be beneficial to prevention and treatment of HCC, especially for obese patients.

Former studies always ignore KPNA2’s transcripts and take them as a whole, which might skip the genetics background and potential functions. In our analysis, the two full-length KPNA2 transcripts demonstrated independently prognostic significance for HCC OS in a paradoxical way. ENST00000330459 of 1977bp with a predominant proportion is unfavorable for HCC outcome while ENST00000537025 of 2456bp with a longer 5′  leading sequence UTR shows an inverse effect. One of the potential explanations is that the two transcripts coding the same protein might have been experiencing competition during evolution and the longer 5′  UTR region has more opportunities to be modified and degraded. Among the KPNA2 protein variants, the canonical full-length KPNA2 protein (UniProt_P52292) is significantly upregulated in HCC nuclei but is rarely presented in nuclei of non-tumor liver tissues. The abnormal nuclear accumulation of P52292 protein indicates an imbalance in nuclear transport cycle and such changes are associated with carcinogenesis and poor prognosis in a multitude of cancers (Stelma et al., 2016). In contrast, the truncated KPNA2 isoforms which lack the NLS-binding domain but retain their ability to bind to importin β exist in nuclei of non-tumor liver tissues and in cytoplasm/membrane of HCC without obvious nuclear translocation, indicating their physiological functions in normal liver cells and failure to transport cargo proteins into nuclei of HCC cells. A previous study evidenced that a truncated KPNA2 lacking the NLS-binding domain predominantly localized in the cytoplasm of breast cancer cells and inhibited nuclear import of p53 (Kim et al., 2000). One possible explanation is that the fragmental KPNA2 variants could affect the full-length KPNA2’s function through competitively binding to importin β. Although detailed functions of the fragmental KPNA2 variants remain unanswered, it is reasonable to speculate that they might be involved in HCC progression by causing inefficient nuclear import of tumor suppressor nuclear proteins.

At last, plasma KPNA2 between HCC patients and healthy groups was compared by ELISA and exhibited lower level with statistical significance in the HCC patients than in the healthy individuals. It was inconsistent with a previous report that serum KPNA2 level was elevated in non-small-cell lung carcinoma patients than in healthy individuals (Wang et al., 2011). Different tumor type is a possible explanation; nuclear translocation of KPNA2 in HCC cells resulting in scarce cytoplasm/membrane expression followed by reduced exocytosis is another reasonable interpretation. Moreover, KPNA2 expression was lower in noncancerous hepatic tissues from most of the HCC patients with or without detectable metastases than in the disease-free normal livers in a GEO dataset GDS3091 (Budhu et al., 2006), implying that the decreased KPNA2 level surrounding the primary HCC lesions would result in its lower concentration in the peripheral blood. However, it’s worth noting that KPNA2 is expressed in most tissues and organs besides the liver. We examined the plasma KPNA2 in HCC patients and normal individuals given that their other tissues and organs were comparable except the difference in their livers between the two groups. If other tissues and organs weigh a lot in the distinction of KPNA2 release, the plasma KPNA2 might not accurately indicate the change of KPNA2 in HCC. It needs to be further verified in future.

Conclusions

In summary, we explored the molecular characteristics and dysregulation of KPNA2 through in-depth multi-omics analysis. The CNV and methylation level of KPNA2 gene varied during HCC development and predicted the disease outcome. The effectiveness of the risk models for HCC OS and DFS prediction presented the advantage of multi-omics analyses of a specific gene. Furthermore, dissection of KPNA2 transcript variants and protein isoforms broadened KPNA2 clinicopathological features. The role of KPNA2 in HCC immunoregulation was highlighted, providing information of immune cell constitutions within HCC microenvironment and for prediction of HCC immune response, and constituting a potential resource in anti-tumor immunotherapy. The negative correlations between KPNA2 and most genes in fatty acid metabolism pathway, its positive relationship with fatty acid synthase activity, and low-fat diet effect on KPNA2 decrease in liver indicated the importance of diet control in HCC prevention and treatment. Taken together, the integrative analyses of KPNA2 provided in-depth knowledge of the molecular mechanisms of HCC pathogenesis and explored promising molecular markers and signatures for HCC prognosis and therapy.

Supplemental Information

Supplemental Information 1 Pre-analysis of KPNA2 in HCC

Pre-analysis of KPNA2 in HCC.

Click here for additional data file.

Supplemental Information 2 Supplementary tables and figures

Click here for additional data file.

Supplemental Information 3 KPNA2-correlated genes in HCC

Spearman correlation nanlysis was used. Only the genes with —correlation coefficient—>0.3 were included.

Click here for additional data file.

Supplemental Information 4 Plasma KPNA2(ng/ml) in HCC and normal controls

Click here for additional data file.

We thank Dr. JM Zeng from University of Macau and his team for their help in the bioinformatics analysis.

Additional Information and Declarations

Competing Interests

Author Contributions

Human Ethics

Data Availability

The authors declare there are no competing interests.

Jinzhong Zhang conceived and designed the experiments, authored or reviewed drafts of the paper, and approved the final draft.

Xiuzhi Zhang conceived and designed the experiments, analyzed the data, prepared figures and/or tables, authored or reviewed drafts of the paper, and approved the final draft.

Lingxiao Wang and Ningning Li performed the experiments, prepared figures and/or tables, and approved the final draft.

Chunyan Kang performed the experiments, analyzed the data, prepared figures and/or tables, and approved the final draft.

Zhefeng Xiao conceived and designed the experiments, performed the experiments, authored or reviewed drafts of the paper, and approved the final draft.

Liping Dai analyzed the data, prepared figures and/or tables, authored or reviewed drafts of the paper, and approved the final draft.

The following information was supplied relating to ethical approvals (i.e., approving body and any reference numbers):

The work involving the plasma specimens was reviewed and approved by the Ethics Committee of Xiangya Hospital of Central South University (201801002).

The following information was supplied regarding data availability:

The raw data of plasma KPNA2 are available in the Supplementary File.

The other data that support the findings of this study are available at:

- TCGA-LIHC dataset, https://portal.gdc.cancer.gov/projects/TCGA-LIHC.

- UCSC: TCGA-LIHC dataset, https://xenabrowser.net/datapages/?cohort=TCGA.

- HPA (https://www.proteinatlas.org/),

- GEO: GSE7117. https://www.ncbi.nlm.nih.gov/geo/query/acc.cgi?acc=GSE7117.

- KEGG: Fatty acid degradation - Homo sapiens (human) (https://www.genome.jp/kegg/pathway/hsa/hsa00071.html).

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
