# Peer review of "Multiomics-based analyses of KPNA2 highlight its multiple potentials in hepatocellular carcinoma"

_PeerJ, doi:10.7717/peerj.12197_

## Round 0.1 · original submission · Minor Revisions

Reviewers have now commented on your paper. You will see that they are advising you of a minor revision of your manuscript. If you are prepared to undertake the work required, I would be pleased to reconsider my decision.

For your guidance, reviewers' comments are appended below.

If you decide to revise the work, please submit a list of changes or a rebuttal against each point that is being raised when you submit the revised manuscript.

·

Basic reporting

Zhang et.al used multi-omics approaches to make the prediction of HCC outcome and presented valuable information regarding the change of KPNA2 variants during HCC. The paper is generally well written and structured. However, the paper has some shortcomings regarding to several data analyses and text:

1). The font size is too small for most of the figures.

2). Figure 5A, Are there any sub-clusters in your PPI analysis? What are your clustering coefficient and the minimum required interaction score you used? Could you indicate in the figure legend what the colors of each dot represent?

3). The finding of the connection of KPNA2 and immune cells is very interesting. Can you find more data to indicate the association of KPNA2 and immune cells in different disease stages? It will be helpful if the authors can provide some experimental information on it.

4). Can you do the co-expressed analysis of KPNA2 to see the gene expression pattern related to HCC? And find out the top positive and negative correlated genes with KPAN2 in the HCC?

5). Figure 8, please explain why using paired samples test?

6). Figure 11. Can the authors also look at the relationship between different KPNA2 transcripts with some HCC’s serum markers, like AFP?

Experimental design

no comment

Validity of the findings

no comment

·

Basic reporting

In the current study, the authors provide a relatively comprehensive analysis of molecular features of KPNA2 in hepatocellular carcinoma (HCC) by using multi-omics-based analyses. This suggests a potential association between KPNA2 and the possibility of using molecular features of KPNA2 as diagnostic and prognostic prediction tools for HCC.

The whole study was well conducted and the manuscript is well written. However, there are still some questions/problems that are needed to be resolved before final acceptance.

1. What are the potential molecular mechanisms that KPNA2 may regulate the genesis and development of HCC? The authors should provide a more detailed introduction about this in the Introduction session based on current knowledge.

2. Please move the data in the supplementary figure 1 to figure 1 because it is critical to the main story. The same action should be performed for the data in figure S4.

3. In the discussion, the authors also describe that “KPNA2 could affect carcinogenesis by mistaken translocation of cancer-associated cargo proteins such as c-Myc and PLAG1, which could be found in the liver-specific KPNA2 PPI network.” Please provide reference to support that KPNA2 could affect carcinogenesis by mistakenly translocating c-myc.

4. Please add the reference for the sentence “Liver is a central immunomodulator that ensures organ and systemic protection while maintaining immunotolerance.”

5. Please re-check the whole manuscript to resolve the grammatical problems. If possible, please rephrase some of the long sentences into short ones, which can make the description easier for the readers to understand.

Experimental design

1. Based on the current study, KPNA2 seems to play its role in the hepatic cells by transporting a variety of proteins between the cell nucleus and cytoplasm. However, at the last of the study, the authors examined KPNA2 in the plasma. Are there any studies suggesting plasmic KPNA2 primarily comes from the liver? Is it possible that the plasma KPNA2 may be released by other cells, e.g. immune cells or from other organs? If yes, this data may not be suitable to indicate the alteration of HCC or the change of KPNA2 in HCC.

Validity of the findings

1. In the discussion, the authors say that “the constitutive risk models involving CpG sites methylation are presumably more predictive for HCC prognosis than KPNA2 expression level alone.” Please provide the evidence to support this conclusion.

2. Interestingly, the authors discussed that KPNA2 could affect carcinogenesis by mistakenly translocating cancer-associated cargo proteins, while a truncated KPNA2 may inhibit the nuclear import of p53. Are there any studies showing how KPNA2 distinctly regulates the transport of oncogenes and tumor suppressor genes?

---

## Round 0.2 · Minor Revisions

Hi, most of the reviewer's questions were well addressed in the revised version. One of the reviewers raised more discussions, please address the reviewer's further questions before we could accept it for publication.

·

Basic reporting

I would like to thank the authors for addressing my initial comments. The authors have provided a nicely detailed and thorough response to the comments from the previous review and have addressed my major concerns. The manuscript now reads with greater focus and clarity. I would like to recommend this manuscript to be published on the journal.

Experimental design

None

Validity of the findings

None

Additional comments

None

·

Basic reporting

I appreciate the authors for providing a thorough rebuttal letter. Based on the replies, I have two further questions.

1. Based on the current study, KPNA2 seems to play its role in the hepatic cells by transporting a variety of proteins between the cell nucleus and cytoplasm. However, at the last of the study, the authors examined KPNA2 in the plasma. Are there any studies suggesting plasmic KPNA2 primarily comes from the liver? Is it possible that the plasma KPNA2 may be released by other cells, e.g. immune cells or from other organs? If yes, this data may not be suitable to indicate the alteration of HCC or the change of KPNA2 in HCC.

Authors’ reply: -Thank you for your reasonable and thoughtful suggestion. KPNA2 is indeed expressed in most normal tissues, up-regulated in a variety of organ originated cancers, and is a plasma protein. We examined the plasma KPNA2 level in HCC patients and normal individuals given that their most other tissues and organs were comparable except the difference of livers between the two groups. It has to be admitted that if other tissues and organs weigh a lot in KPNA2 release, the plasma KPNA2 could not accurately reflect the change of KPNA2 in HCC. It needs to be further verified in future. Thus, we include it as another potential explanation for the ELISA outcome in the Discussion section. (Line: 452-456)

Q: Can the authors provide evidence to support that “most other tissues and organs were comparable except the difference of livers between the two groups”?

2. Interestingly, the authors discussed that KPNA2 could affect carcinogenesis by mistakenly translocating cancer-associated cargo proteins, while a truncated KPNA2 may inhibit the nuclear import of p53. Are there any studies showing how KPNA2 distinctly regulates the transport of oncogenes and tumor suppressor genes?

Authors’ reply: -Most studies focused on identifying the cargo proteins of which KPNA2 mediated nuclear localization. Both oncoproteins and tumor suppressor proteins are nuclear translocated with KPNA2 such as E2F1, OCT4, c-Myc, p53, p27, MDC1, FGF1/2, LEF-1, CHK1, BRCA1, S100A2, S100A6, RECQL, RAC1, p65, JNK1, STAT3, c-Jun, NBS1, TBP-2 (Han & Wang 2020; Martinez-Olivera et al. 2018; Zhou et al. 2021). Then, their downstream signaling including PI3K/AKT, P53, MAPK, cell cycle, Wnt/β-catenin, and EMT-related pathways affect the growth, invasion, and metastasis of tumors (Han & Wang 2020). Given that KPNA2 binds to oncoproteins and tumor suppressor proteins indistinctively, it might depend on oncogenic or tumor suppressive signaling to distinctively regulate the transport of oncoproteins and tumor suppressor proteins. For example, KPNA2 was one of candidate target genes of p53 which inhibits KPNA2 in liver cancer to suppress tumor cell growth. Moreover, lncRNAs, microRNAs, and transcriptional factors such as E2F1 (Drucker et al. 2019) have been reported to regulate KPNA2 expression.

Q: Could the authors add their reply for this question into the discussion part?

Experimental design

N/A

Validity of the findings

N/A

---

## Round 0.3 · accepted · Accept

Congratulations, all reviewer's concerns are well addressed, and the current version is good for publication. The KPNA2 related genome changes, mRNA and protein expression alterations, as well as the affected pathways are all consistent, making this paper a strong candidate for publication.

·

Basic reporting

All my questions were satisfactorily answered by the authors.

Experimental design

N/A

Validity of the findings

N/A

Additional comments

N/A